# Is Urinary Netrin-1 a Good Marker of Tubular Damage in Preterm Newborns?

**DOI:** 10.3390/jcm10040847

**Published:** 2021-02-19

**Authors:** Monika Kamianowska, Marek Szczepański, Natalia Chomontowska, Justyna Trochim, Anna Wasilewska

**Affiliations:** 1Department of Neonatology and Neonatal Intensive Care, Medical University of Bialystok, 15-276 Bialystok, Poland; szczepanski5@gazeta.pl (M.S.); natkachom@tlen.pl (N.C.); 2Department of Pediatric Laboratory Diagnostics, Medical University of Bialystok, 15-276 Bialystok, Poland; justynatrochim@wp.pl; 3Department of Pediatrics and Nephrology, Medical University of Bialystok, 15-276 Bialystok, Poland; annwasil@interia.pl

**Keywords:** netrin-1, renal tubular damage, premature newborns

## Abstract

There is a lack of a good marker for early kidney injury in premature newborns. In recent publications, netrin-1 seems to be a promising biomarker of kidney damage in different pathological states. The study aimed to measure the urinary level of netrin-1 depending on gestational age. A prospective study involved 88 newborns (I-60 premature newborns, II-28 healthy term newborns). Additionally, premature babies were divided for 2 groups: IA-28 babies born between 30–34 weeks of gestation and IB-32 born at 35–36 weeks. The median urinary concentration of netrin-1 was: IA-(median, Q1–Q3) 63.65 (56.57–79.92) pg/dL, IB-61.90 (58.84–67.17) pg/dL, and II-60.37 (53.77–68.75) pg/dL, respectively. However urinary netrin-1 normalized by urinary concentration of creatinine were IA-547.9 (360.2–687.5) ng/mg cr., IB-163.64 (119.15–295.96) ng/mg cr., and II-81.37 (56.84–138.58) ng/mg cr., respectively and differ significantly between the examined groups (*p* = 0.00). The netrin-1/creatinine ratio is increased in premature babies. Further studies examining the potential factors influencing kidney function are necessary to confirm its potential value in the diagnosis of subclinical kidney damage in premature newborns.

## 1. Key Notes

Premature babies are exposed to kidney dysfunction. There is a lack of a good marker for early kidney injury in this group of children. In recent publications, netrin-1 seems to be a promising biomarker of kidney damage in different pathological states. We showed that the netrin-1/creatinine ratio is increased in premature babies which may have potential diagnostic value in the diagnosis of subclinical kidney damage in premature newborns.

## 2. Introduction

Worldwide data show that about 15 million premature babies are born each year [1]. The significant improvement in intensive care of newborns has led to increased survival rate of premature babies [1,2]. Unfortunately, although the survival of the smallest children has improved, long-term morbidity remains high [3]. It is believed that renal dysfunction in preterm babies promises poor short- and long-term results, regardless of concomitant diseases and interventions, both in children and adults [4,5]. This is because the nephrogenesis lasts from 6 to 36 weeks of gestation, and almost 60% of nephrons are formed only in the third trimester of pregnancy [6].

It is known that the primary cause of complications of early and late prematurity is immaturity. However, early detection and implementation of treatment in the event of perinatal problems can significantly reduce late multi-organ damage [7]. Owing to the fact that the traditional markers of kidney damage (filtration marker-creatinine and markers of kidney damage, such as urine sediment abnormalities and albuminuria) are not very precise (they increase with delay after the operation of the damage, are not specific to determine the site of damage, depend on factors not related to kidney damage), new, more effective biomarkers are being sought [8,9]. 

Netrin-1 a 72 KDa laminin-related secreted protein, initially found in the central nervous system during neurogenesis is expressed in many tissues, including renal tissues. It is shown that netrin-1 promotes cell migration, angiogenesis, tissue morphogenesis, and takes a significant part in the regulation of the inflammatory process [10]. It is unlikely that netrin-1 is filtered by the glomerulus under basic conditions due to its molecular weight, however, in acute and chronic kidney damage, netrin-1 is highly induced and excreted to the urine of both animals and humans [11,12]. This property caused recognition of netrin-1 as an early diagnostic marker of kidney damage [13]. 

In this study, we analyzed whether prematurity affects the function of renal tubules assessed by the tubular damage marker-netrin-1.

## 3. Patients and Methods

### 3.1. Patient Recruitment and Sample Collection

This prospective study included 88 neonates hospitalized at the Department of Neonatology and Intensive Neonatal Care, Medical University of Bialystok, Poland between December 2017 and December 2018. Sixty newborns were born prematurely between 30–36 weeks of gestation. All these newborns were appropriate for gestational age (AGA) with the weight between the 10th and 90th percentile of birth weight for their gestational age using normalized growth curves [14]. Their clinical condition was assessed as good or fair. Since the premature babies born between 30–36 weeks of gestation constitute a very diverse group of children, in terms of both body weight and clinical status we identified two groups: IA-28 newborns born between 30–34 weeks and IB-32 newborns born between 35–36 weeks of gestation. Twenty-eight healthy newborns (reference group) were the result of normal pregnancies without any prenatal and perinatal complications. Subjects enrolled in this study met the following criteria: normal prenatal and postnatal ultrasound examination of the kidney, and good or average clinical condition. The exclusion criteria from the study included Apgar (Appearance, Pulse, Grimace, Activity, Respiration) score lower than 4 in the 1st minute after birth, any congenital anomaly, severe clinical condition, inborn error of metabolism, kidney damage, heart disease, abnormal ultrasound examination of the kidney, abnormal to a large extent the image of the central nervous system (accepted hyperechoic zones around the lateral ventricles of the brain and intraventricular hemorrhage grade I and II), elevated inflammatory markers, abnormal laboratory tests, use of mechanical ventilation or drugs (antibiotics, diuretics, catecholamines). The children of mothers with a burdened medical history were excluded from the study too. 

In the study group, urine samples were collected using single-use sterile bags (Medres, Zabrze, Poland). Urine was collected once in the first or second day of life. The blood samples were collected during routine practice in the Unit during the first or second day of life. We used S-Monovette 1.2 mL, Clotting Activator/Serum test tubes (Sarstedt AG & Co, Nümbrecht, Germany) to venous blood sampling. Blood cell morphology tests and blood biochemistry tests were performed immediately after taking blood samples. The urine samples obtained after centrifugation were kept in the fridge (4 °C) for maximum of 2 h and then frozen at −80 °C. We did not use repeated freeze-thaw cycles. 

### 3.2. Determination of Urinary Netrin-1

The levels of urinary netrin-1 were measured using a commercially available ELISA kit (Cloud—Clone Corp. R&D Systems, Katy, TX, USA) according to manufacturer’s instructions and were expressed in picograms per milliliter. According to specifications of this kit the detection range was 31.2–2000 pg/mL, the minimum detectable dose of netrin-1 was typically less than 12.4 pg/mL. The mean intra-assay coefficients of variation (CV) for urine netrin-1 were <10%, and inter-assay coefficients were <12%.

The urinary concentrations of netrin-1 were normalized for the urinary concentrations of creatinine determined with Jaffé’s method to account for the potential confounding effects of urinary dilution and expressed in nanograms per milligram creatinine (ng/mg cr). 

The morphology and serum biochemistry tests were performed in the Department of Laboratory Diagnostics University Clinical Hospital in Bialystok.

Estimated GFR (glomerular filtration rate) was calculated according to Schwartz formula for the term babies (eGFR = 0.45 × L/Scr., and preterm babies eGFR = 0.33 × L/Scr where L is the length in centimeters, and Scr is serum creatinine in milligrams per deciliter). 

## 4. Statistical Analysis 

The results were analyzed with Statistica 13.3 package (StatSoft, Cracow, Poland). Discrete variables were expressed as counts (percentage), continuous variables as median and quartiles (Q1–Q3). The Shapiro -Wilk test was used to determine normal distribution. Because the data were not normally distributed, Mann-Whitney U-test was used for intergroup comparisons of continuous variables. Spearman’s rank correlation coefficients were used to determine the direction and power of association between the urinary netrin-1/cr. ratio and other variables. The results were considered significant at *p* < 0.05. 

## 5. Results

The study included 88 neonates: sixty babies born prematurely between 30–36 weeks of pregnancy and twenty-eight healthy newborns. Both groups were sex-matched (*p* > 0.05). Table 1 shows the characteristics of the premature babies. 

In younger babies, parameters of physical development (birth weight, length, head and chest circuit) were significantly lower than in term newborns. All these newborns, however, were appropriate for gestational age and there were no statistically significant differences, even if the children were divided into 10–50 percentile and 51–90 percentile subgroups. Younger babies had lower Apgar scores at 1st and 3rd minute. All newborns had Apgar scores ≥ 8 at a 5th and 10th minute. No statistically significant differences were found in the delivery of premature babies in both examined groups.

Blood morphology and biochemical tests did not show any deviations from the norm in both groups of preterm children. Younger newborns had statistically significantly higher leukocytosis, urea and alanine aminotransferase concentrations (Table 2).

Renal function parameters such as serum and urinary concentration of creatinine, estimated GFR and urine output were normal in all babies. There was no statistically significant difference in serum concentration of creatinine between all children. The urine concentration of creatinine was significantly higher in older premature newborns and highest in term newborns (*p* < 0.001). The eGFR was similar in premature babies and statistically significantly higher in term newborns (Table 3). 

The urinary concentration of netrin-1 did not differ between all babies. However, urinary netrin-1 normalized by urinary creatinine differed significantly between all the groups. The highest levels of netrin-1/cr. were observed in babies born between 30–34 weeks of gestation. Also, in babies born between 35–36 weeks of gestation, urinary level of netrin-1/cr. was higher comparing to the reference group (*p* < 0.001; *p* < 0.001), respectively (Table 3). 

We did not find any relationship between the concentration of urinary netrin-1 and also urinary netrin-1/cr. and gender, way of delivery, Apgar score, prenatal steroid therapy, respiratory disorders (use of nasal continuous positive airway pressure (nCPAP), oxygen therapy), parenteral nutrition.

## 6. Discussion

Detecting kidney damage early and with the precise location is currently a priority in the diagnosis of kidney diseases. It allows to quickly implement the appropriate treatment, monitor its effectiveness and prognosis, and due to that increase, the chance for cure [8]. Netrin-1 can be an early marker of tubular kidney damage [13,15,16]. Ramesh et al. showed that urinary netrin-1 excretion in children increased at 2 h after cardiopulmonary bypass (CPB), peaked at 6 h and remained elevated up to 48 h after CPB. The 6-h urine netrin-1 measurement strongly correlated with duration and severity of AKI (acute kidney injury), as well as length of hospital stay (all *p* < 0.05) [15]. Higher concentrations of urinary netrin-1 have been found by Jayakumar et al. in normoalbuminuric diabetic patients than in the healthy groups and even more elevated in patients diagnosed with microalbuminuria or overt nephropathy [13]. There are only a few published studies on the determination of netrin-1 in newborns [8,17,18]. 

This prospective study was the first conducted to determine the values of urinary netrin-1 in premature newborns in good or stable clinical condition. In this study, we hypothesized that the prematurity could lead to renal tubular injury and urinary netrin-1 could be used as an early marker of tubular damage. It is known that there are several factors that can lead to tubular damage. We have therefore identified a group of 60 newborns with birth weight appropriate for gestational age and in clinical condition assessed as good or stable. None of the children required mechanical ventilation and drugs. They all had normal prenatal and postnatal ultrasound examination of the kidney and did not have any deviation in laboratory tests and parameters of renal function (Table 2 and Table 3). In our studied groups, eGFR was comparable in premature babies and statistically significantly higher in term newborns, which is in agreement with the literature [19]. The urine concentration of creatinine was the lowest in the babies from the group born between 30–34 weeks of pregnancy. This is in line with the results of Al-Dahhan et al. who similarly showed that urinary excretion of creatinine positively correlates with body weight and gestational age [20].

We showed that the median of urinary netrin-1 level did not differ between the examined groups. However, it is well-known that the values of urine parameters closely depend on water content or urinary concentration throughout the day, we used urine creatinine concentration to normalize the results [8]. It turned out that we found the highest urinary level of netrin-1/cr. in babies born between 30–34 weeks of gestation and it differs significantly from the other two groups. Also, in babies born between 35-36 weeks of gestation, urinary level of netrin-1/cr. was higher when compared to the reference group (*p* = 0.00; *p* = 0.00), respectively (Table 3). Al Morsy et al. in their study showed higher netrin level-1 in premature babies with AKI-880.10 ± 69.12 pg/mL than in premature babies without AKI-693.10 ± 47.15 pg/mL. They did not normalize netrin-1 concentration by urine creatinine. The results they obtained were higher than those obtained by us. However, the sample size in this study was limited (20), children were younger (32.2 ± 1.24 wk), born with definitely lower Apgar score (6 (6–7), at 5th minutes), all required mechanical ventilation or nCPAP respiratory support [17]. To the best of our knowledge, there are very few studies reporting urinary netrin-1 concentrations in newborns or children [16,18,21]. In the study by Oncel et al. the concentrations of urinary netrin-1 on the first postnatal day were higher in newborns with perinatal asphyxia (848 ± 239 pg/mL; mean ± SD) compared to controls (592 ± 181 pg/mL mean ± SD) [21]. Unfortunately, urinary netrin-1 was not normalized by urinary creatinine in this study. Cao et al. show that the newborns with asphyxia had significantly higher urinary levels of netrin-1 within 48 h after birth (*p* < 0.05) [18]. Övünç Hacıhamdioğlu et al. found that obese patients had significantly higher netrin-1 excretion than the controls (841.68 ± 673.17 vs. 228.94 ± 137.25 pg/mg cr., *p* = 0.000; mean ± SD) [16]. 

In the further analysis, we paid attention to other factors that might affect the function of the renal tubules. We did not find any relationship between the concentration of urinary netrin-1 or urinary netrin-1/cr. and gender, delivery, birth weight, percentile of birth weight, Apgar score, prenatal steroid therapy, respiratory disorders (use of nCPAP, oxygen therapy), parenteral nutrition. 

Considering the results of a study by Dakouane-Giudicelli et al. and Kang et al. who confirmed that netrin-1 plays a significant role in neurogenesis and angiogenesis, being a significant factor into the establishment of neurovascular networks in the developing kidney we suggest that the results of our study also show on the rule of netrin-1 as a marker of maturation of the kidneys in premature babies [22,23].

To the best of our knowledge, this is the first work suggesting that urinary netrin-1 normalized by urinary creatinine is a good marker of subclinical tubular damage in preterm newborns. It should be noted that this is a preliminary observation that should be confirmed in a multicenter study. Besides, it would be worthwhile to conduct a long-term study to analyze the netrin-1/cr value, especially in correlation with eGFR, in further stages of life. Because we are unable to exclude the influence of the kidney maturation process on netrin-1/cr value, it seems necessary to also analyze this value in premature babies in serious clinical condition.

## Figures and Tables

**Table 1 jcm-10-00847-t001:** Characteristics of premature newborn.

Parameters	Premature (I) (30–36 wk) (*n* = 60)	Premature (IA) (30–34 wk) (*n* = 28)	Premature (IB) (35–36 wk) (*n* = 32)	*p*
	Median (Q1–Q3)			
Gender (boys/girls)	33/27	16/12	17/15	NS
Gestational age (weeks)	35 (33–36)	33 (32–34)	36 (35–36)	<0.001
Vaginal delivery/Caesarean delivery	16/44	10/22	6/22	NS
Birth weight (g)	2450 (2195–2740)	2295 (1720–2450)	2620 (2415–2800)	<0.001
Birth weight (10–50 percentile/ 51–90 percentile)	17/43	7/21	10/22	NS
Length (cm)	50 (47–52)	48.00 (45–50)	52 (49.5–53)	<0.001
Head circuit (cm)	32 (31–33.5)	31 (29–32)	33 (32–34)	<0.001
Chest circuit (cm)	30 (28–31)	28.5 (26.5–30)	31 (30–32)	<0.001
1 min Apgar score (8–10/4–7)	39/21	13/15	26/6	0.02
3 min Apgar score (8–10/4–7)	47/13	18/10	29/3	0.01
5 min Apgar score (8–10/4–7)	60/0	28/0	32/0	NS
10 min Apgar score (8–10/4–7)	60/0	28/0	32/0	NS
nCPAP	18	17	1	<0.001
Oxygen therapy	24	19	5	<0.001
Parenteral nutrition	29	24	5	<0.001
Prenatal steroid therapy	12	12	0	<0.001

*p*-30–34 wk and 35–36 wk; NS non statistical.

**Table 2 jcm-10-00847-t002:** The results of basic laboratory tests of premature newborns.

Parameters	Premature (I) (30–36 wk) (*n* = 60)	Premature (IA) (30–34 wk) (*n* = 28)	Premature (IB) (35–36 wk) (*n* = 32)	*p*
Leukocytes × 10^3^/µL	14.65 (11.57–17.2)	11.57 (10.04–15.58)	16.53 (14.02–18.92)	0.001
Hemoglobin (g/dL)	17.95 (16.7–18.95)	17.75 (15.6–19.05)	18.11 (16.9–18.95)	NS
Hematocrite (%)	49.95 (46.6–52.51)	49.20 (42.55–52.75)	50.1 (47.4–52.4)	NS
Platelets × 10^3^/µL	258 (210–293.5)	267 (232–300.5)	247.5 (204–290.5)	NS
Urea (mg/dL)	25.5 (18–32)	29.50 (22–45.5)	22.5 (15–29.5)	<0.001
Aspartate aminotransferase (IU/L)	48.5 (38–59.5)	40.50 (34.5–59)	22.5 (15–29.5)	NS
Alanine aminotransferase (IU/L)	10 (7–13)	8.00 (6–11)	11.5 (9–14.5)	0.001
Bilirubin (mg/dL)	5.19 (4.05–6.2)	5.45 (3.9–6.25)	4.8 (4.1–6.13)	NS
Protein (mg/dL)	4.85 (4.4–5.2)	4.85 (4.55–5.2)	4.85 (4.3–5.2)	NS
Sodium (mmol/L)	141.5 (139–143)	141 (138–143)	141 (140–143)	NS
Potassium (mmol/L)	4.97 (4.5–5.48)	4.89 (4.5–5.47)	5.07 (4.47–5.48)	NS
Calcium (mmol/L)	2.12 (2.02–2.20)	2.14 (2.03–2.22)	2.12 (2.01–2.18)	NS
Magnesium (mmol/L)	0.86 (0.81–0.92)	0.86 (0.81–0.96)	0.84 (0.81–0.91)	NS
Phosphorus (mmol/L)	2.01 (1.64–2.28)	0.86 (0.81–0.96)	0.84 (0.81–0.91)	NS

*p*-30–34 wk and 35–36 wk.

**Table 3 jcm-10-00847-t003:** Median values and interquartile range (IQR) of examined parameters in premature newborns and the reference group.

Parameters	Premature (I) (30–36wk) (*n* = 60)	Premature (IA) (30–34 wk) (*n* = 28)	Premature (IB) (35–36wk) (*n* = 32)	Reference group (II) (*n* = 28)	*p* _1_	*p* _2_	*p* _3_	*p* _4_
	Median (Q1–Q3)							
Netrin-1 (pg/mL)	62.68 (57.70–75.19)	63.65 (56.57–79.92)	61.90 (58.84–67.17)	60.37 (53.77–68.75)	NS	NS	NS	NS
Netrin-1/creatinine (ng/mg cr.)	285.41 (142.55–543.71)	543.71 (294.38–682.78)	163.64 (119.15–295.96)	81.37 (56.84–138.58)	<0.001	<0.001	<0.001	<0.001
Urine creatinine (mg/dL)	23.88 (3.43–43.58)	14.95 (9.47–25.05)	35.26 (21.38–57.73)	82.315 (38.99–118.66)	<0.001	<0.001	<0.001	<0.001
Serum creatinine (mg/dL)	0.66 (0.61–0.75)	0.635 (0.59–0.68)	0.7 (0.63–0.76)	0.68 (0.55–0.80)	NS	NS	NS	NS
eGFR (mL/min/1.73 m^2^)	24.58 (21.71–27.11)	24.38 (21.45–27.82)	24.86 (21.99–26.60)	37.62 (33.75–47.97)	<0.001	<0.001	<0.001	NS

*p*_1_-30–36 wk and reference group; *p*_2_-30–34 wk and reference group; *p*_3_-35–36 wk and reference group; *p*_4_-30–34 wk and 35–36 wk creatinine; eGFR estimated glomerular filtration rate.

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
