# Peer review of "Is Urinary Netrin-1 a Good Marker of Tubular Damage in Preterm Newborns?"

_jcm, 2021, doi:10.3390/jcm10040847_

Round 1
Reviewer 1 Report
The Authors aim to asses if prematurity affects the function of renal tubules and they use netrin-1 as a marker of tubular damage. They justify the choice of this marker citing studies in mice and in adults and children, mostly affected by diabetes.
The design of the study is prospective and three groups of patients are compared: newborns between 30 and 34 weeks, newborns between 35 and 36 weeks and healthy newborns as control group. Premature newborns with other possible diseases were excluded. Thus the comparison is between "healthy" premature newborns at different gestational ages and healthy newborns. The outcome is the level of netrin-1/creatinine in the three groups. The statistical analysis is adequate.
The main result shows that the ratio netrin-1/creatinine is higher in the early pre-term group.
The Authors conclude that, since the ratio netrin-1/creatinine is a marker of tubular damage, pre-term newborns may have a tubular damage.
Some questions arise about the sensitivity and specificity of netrin-1 as marker of tubular damage, a marker suggested only by observational studies, not addressed to demonstrate its validity as diagnostic tool. The Authors may only have observed a process of maturation of the kidneys at different stages and the excretion of netrin-1 may be a marker of this process, not a sign of disease. I think that this point must be argumented and included in the discussion. After all, the Authors show that the excretion of netrin-1 standardised per creatininuria is different according to the gestational age, the rest is only hypothesis.
Author Response
Monika Kamianowska, Assoc. Prof Białystok, 08th of February 2021
Medical University in Białystok
Department of Neonatology and Neonatal Intensive Care Unit
- C. Skłodowskiej 24a St.
15-276 Białystok
POLAND
" Journal of Clinical Medicine"
Ms. Flora Diao
Assistant Editor
Dear
With reference to your letter 04th of February 2021 I enclosed a revised manuscript: " Is urinary netrin-1 a good marker of tubular damage in pre-term newborns?" written by Monika Kamianowska, Marek Szczepański, Natalia Chomontowska, Justyna Trochim, Anna Wasilewska, for publishing consideration. The work was done in the Department of Neonatology and Neonatal Intensive Care, Medical University of Bialystok, Poland.
I enclose the respond in a point-by point fashion to the referees’ comments and I send it as a new manuscript.
I thank you in advance for the time and effort you expend considering my work.
Sincerely Yours
Monika Kamianowska
According to the Reviewer 1
The Authors aim to asses if prematurity affects the function of renal tubules and they use netrin-1 as a marker of tubular damage. They justify the choice of this marker citing studies in mice and in adults and children, mostly affected by diabetes.
The design of the study is prospective and three groups of patients are compared: newborns between 30 and 34 weeks, newborns between 35 and 36 weeks and healthy newborns as control group. Premature newborns with other possible diseases were excluded. Thus the comparison is between "healthy" premature newborns at different gestational ages and healthy newborns. The outcome is the level of netrin-1/creatinine in the three groups. The statistical analysis is adequate.
The main result shows that the ratio netrin-1/creatinine is higher in the early pre-term group.
The Authors conclude that, since the ratio netrin-1/creatinine is a marker of tubular damage, pre-term newborns may have a tubular damage.
Some questions arise about the sensitivity and specificity of netrin-1 as marker of tubular damage, a marker suggested only by observational studies, not addressed to demonstrate its validity as diagnostic tool. The Authors may only have observed a process of maturation of the kidneys at different stages and the excretion of netrin-1 may be a marker of this process, not a sign of disease. I think that this point must be argumented and included in the discussion. After all, the Authors show that the excretion of netrin-1 standardised per creatininuria is different according to the gestational age, the rest is only hypothesis.
We agree the results may be indicate for the delayed process of maturation and the excretion of netrin-1is a marker of this process. Further studies in older children born as premature babies should be performed to check if increased level of netrin excretion still exists.
We added according to Reviewer’s suggestion to Discussion:
‘Considering the results of a study by Dakouane-Giudicelli et al. and Kang et al. who confirmed that netrin-1 plays a significant role in neurogenesis and angiogenesis, being a significant factor into the establishment of neurovascular networks in the developing kidney we suggest that the results of our study also show on the rule of netrin-1 as a marker of maturation of the kidneys in premature babies [22,23].’
To the best of our knowledge, this is the first work suggesting that urinary netrin-1 normalised by urinary creatinine is a good marker of subclinical tubular damage in preterm newborns. It should be noted that this is a preliminary observation that should be confirmed in a multicenter study. Besides, it would be very valuable to conduct a long-term study to analyze the netrin-1/cr value especially in correlation with eGFR, in further life further stages of life. Unable to exclude the influence of the kidney maturation process on netrin-1/cr value, it seems necessary to also analyze this values in premature babies in serious clinical condition.
Reviewer 2 Report
Kamianowska et al. conducted a prospective cohort study on premature neonates admitted to the neonatal intensive care unit at Medical University of Bialystok (Poland). A total of 88 infants were included in the study: 60 were born prematurely and the rest were healthy newborns. Authors claimed that this was the first prospective study to test the association of urinary netrin-1 and renal tubular injury in premature infants. The study contains a relatively small number of infants and is conducted at a single center but it is generally well-designed and straightforward. I have the following concerns and suggestions:
1) What stage of life were the urinary and serum samples collected from the infants? It is known that the serum creatinine in the first 2-3 days of life might be a reflection of the mother's values.
2) I have the following concerns regarding the P values:
i) In Tables 1 and 2, there are three groups but only one P values at the last column. What exactly does this P value represent?
ii) Please replace the values ‘0.00’ with either ‘<0.01’ or ‘<0.001’ in Tables 1-3.
3) Please define what APGAR (lane 77), GFR (lane 106), NS (Table 1), AKI (lane 161) stand for when those abbreviations first appear in the text.
4) Please correct the following typos:
- injury in in this group >> injury in this group (lane 30)
- centimetres >> centimeters (lane 107)
- decilitre >> deciliter (lane 108)
- (p=0.00, p=0.00) >> (p=0.00) (lane 137)
- Mediana >> Median (Table 3)
- m2 >> please make ‘2’ superscript (Table 3)
- multicentre >> multicenter (lane 212)
5) Please do the following changes in the related sentences:
- potential diagnostic value potential value in >> potential diagnostic value in (lane 33)
- a new, more effective biomarkers >> new, more effective biomarkers (lane 50)
- Table 1 shows the characteristics of the premature babies (Table 1) >> Table 1 shows the characteristics of the premature babies (lane 120)
- parameters: serum and >> parameters such as serum and (lane 133)
- Detecting early kidney damage >> Detecting kidney damage early (lane 154)
- There are only a few published study >> There are only a few published studies (lane 164)
- significant higher >> significantly higher (lane 175)
- what is in agreement >> which is in agreement (lane 176)
- the difference between both other groups was statistically significant >> it differs significantly from the other two groups (lane 184)
- and also urinary netrin-1/cr. >> or urinary netrin-1/cr. (lane 205)
- further life >> further stages of life (lane 213)
6) Please review/re-write the following sentences; those are either too long and not clear or do not sound correct grammatically:
It allows to implement … (lane 155)
Author Response
Monika Kamianowska, Assoc. Prof Białystok, 08th of February 2021
Medical University in Białystok
Department of Neonatology and Neonatal Intensive Care Unit
- C. Skłodowskiej 24a St.
15-276 Białystok
POLAND
" Journal of Clinical Medicine"
Ms. Flora Diao
Assistant Editor
Dear
With reference to your letter 04th of February 2021 I enclosed a revised manuscript: " Is urinary netrin-1 a good marker of tubular damage in pre-term newborns?" written by Monika Kamianowska, Marek Szczepański, Natalia Chomontowska, Justyna Trochim, Anna Wasilewska, for publishing consideration. The work was done in the Department of Neonatology and Neonatal Intensive Care, Medical University of Bialystok, Poland.
I enclose the respond in a point-by point fashion to the referees’ comments and I send it as a new manuscript.
I thank you in advance for the time and effort you expend considering my work.
Sincerely Yours
Monika Kamianowska
According to the Reviewer 2
Kamianowska et al. conducted a prospective cohort study on premature neonates admitted to the neonatal intensive care unit at Medical University of Bialystok (Poland). A total of 88 infants were included in the study: 60 were born prematurely and the rest were healthy newborns. Authors claimed that this was the first prospective study to test the association of urinary netrin-1 and renal tubular injury in premature infants. The study contains a relatively small number of infants and is conducted at a single center but it is generally well-designed and straightforward. I have the following concerns and suggestions:
1) What stage of life were the urinary and serum samples collected from the infants? It is known that the serum creatinine in the first 2-3 days of life might be a reflection of the mother's values.
Our samples were collected during 1-2nd day. We agree that the serum creatinine concentration during first days may be influenced by mothers value, but I we would like to underline that all the serum samples from the neonates were collected and the same time and the serum level of creatinine did not differ between the groups and subgroup. All mothers of our children were healthy and had normal renal function.
The urinary level of creatinine in the same babies differs significantly between the groups of neonate what it probably affected by renal immaturity and potential injury during the early postnatal period. It is well known that at the time when the majority of preterm infants are born, renal development is still ongoing (Sutherland MR, Gubhaju L, Moore L, Kent AL, Dahlstrom JE, Horne RS, Hoy WE, Bertram JF, Black MJ. Accelerated maturation and abnormal morphology in the preterm neonatal kidney. J Am Soc Nephrol 22: 1365–1374, 2011.) and renal function is accordingly immature (Gubhaju L, Sutherland MR, Black MJ. Preterm birth and the kidney: implications for long-term renal health. Reprod Sci 18: 322–333, 2011.).
I have the following concerns regarding the P values:
Table 3. Median values and interquartile range (IQR) of examined parameters in premature newborns and the reference group.
|
Parameters |
Premature (I) (30-36wk) (n=60) |
Premature (IA) (30-34 wk) (n=28) |
Premature (IB) (35-36wk) (n=32) |
Reference group (II) (n=28) |
p1 |
p2 |
p3 |
p4 |
|
Median (Q1-Q3) |
||||||||
|
Urine creatinine (mg/dl) |
23.88 (3.43-43.58) |
14.95 (9.47-25.05) |
35.26 (21.38-57.73) |
82.315 (38.99-118.66) |
<0.001 |
<0.001 |
<0.001 |
<0.001 |
|
Serum creatinine (mg/dl) |
0.66 (0.61-0.75) |
0.635 (0.59- 0.68) |
0.7 (0.63-0.76) |
0.68 (0.55-0.80) |
NS |
NS |
NS |
NS |
p1 -30-36 wk and reference group; p2- 30-34 wk and reference group; p3-35-36 wk and reference group; p4- 30-34 wk and 35-36 wk creatinine; eGFR estimated glomerular filtration rate.
- i) In Tables 1 and 2, there are three groups but only one P values at the last column. What exactly does this P value represent?
We added according to Reviewer’s suggestion to Results: p-30-34 wk and 35-36 wk
- ii) Please replace the values ‘0.00’ with either ‘<0.01’ or ‘<0.001’ in Tables 1-3.
We changed according to Reviewer’s suggestion in Results: p ‘0.00’ to this one: ‘<0.001’
3) Please define what APGAR (lane 77), GFR (lane 106), NS (Table 1), AKI (lane 161) stand for when those abbreviations first appear in the text.
We changed according to Reviewer’s suggestion in Patients and Methods
‘The exclusion criteria from the study included Apgar score lower than 4 in the 1st minute after birth, any congenital anomaly, severe clinical condition, inborn error of metabolism, kidney damage, heart disease, abnormal ultrasound examination of the kidney, abnormal to a large extent the image of the central nervous system (accepted hyperechoic zones around the lateral ventricles of the brain and intraventricular hemorrhage grade I and II), elevated inflammatory markers, abnormal laboratory tests, use of mechanical ventilation or drugs (antibiotics, diuretics, catecholamines).’
to this one:
‘The exclusion criteria from the study included Apgar (Appearance, Pulse, Grimace, Activity, Respiration) score lower than 4 in the 1st minute after birth, any congenital anomaly, severe clinical condition, inborn error of metabolism, kidney damage, heart disease, abnormal ultrasound examination of the kidney, abnormal to a large extent the image of the central nervous system (accepted hyperechoic zones around the lateral ventricles of the brain and intraventricular hemorrhage grade I and II), elevated inflammatory markers, abnormal laboratory tests, use of mechanical ventilation or drugs (antibiotics, diuretics, catecholamines).’
‘Estimated GFR was calculated according to Schwartz formula for the term babies (eGFR= 0.45 x L/Scr., and preterm babies eGFR= 0.33 x L/Scr where L is the length in centimetres, and Scr is serum creatinine in milligrams per decilitre).’
to this one:
Estimated GFR (glomerular filtration rate) was calculated according to Schwartz formula for the term babies (eGFR= 0.45 x L/Scr., and preterm babies eGFR= 0.33 x L/Scr where L is the length in centimetres, and Scr is serum creatinine in milligrams per decilitre).
We added according to Reviewer’s suggestion to Results:
NS non statistical
We changed according to Reviewer’s suggestion in Discussion
The 6-hour urine netrin-1 measurement strongly correlated with duration and severity of AKI, as well as length of hospital stay (all p < 0.05) [15].
to this one:
The 6-hour urine netrin-1 measurement strongly correlated with duration and severity of AKI (acute kidney injury), as well as length of hospital stay (all p < 0.05) [15].
4) Please correct the following typos:
We corrected according to Reviewer’s suggestion the following typos:
- injury in in this group >> injury in this group (lane 30)
There is a lack of a good marker for early kidney injury in this group of children.
- centimetres >> centimeters (lane 107), - decilitre >> deciliter (lane 108)
Estimated GFR (glomerular filtration rate) was calculated according to Schwartz formula for the term babies (eGFR= 0.45 x L/Scr., and preterm babies eGFR= 0.33 x L/Scr where L is the length in centimeters, and Scr is serum creatinine in milligrams per deciliter).
- (p=0.00, p=0.00) >> (p<0.001) (lane 137)
The urine concentration of creatinine was significantly higher in older premature newborns and highest in term newborns (p<0.001).
- Mediana >> Median (Table 3)
- m2 >> please make ‘2’ superscript (Table 3)
- multicentre >> multicenter (lane 212)
It should be noted that this is a preliminary observation that should be confirmed in a multicenter study.
5) Please do the following changes in the related sentences:
We changed according to Reviewer’s suggestion:
- potential diagnostic value potential value in >> potential diagnostic value in (lane 33)
We showed that the netrin-1/creatinine ratio is increased in premature babies which may have potential diagnostic value in the diagnosis of subclinical kidney damage in premature newborns.
- a new, more effective biomarkers >> new, more effective biomarkers (lane 50)
Owing to the fact that the traditional markers of kidney damage (filtration marker - creatinine and markers of kidney damage, such as urine sediment abnormalities and albuminuria), are not very precise (they increase with delay after the operation of the damage, are not specific to determine the site of damage, depend on factors not related to kidney damage) new, more effective biomarkers are being sought [8,9].
- Table 1 shows the characteristics of the premature babies (Table 1) >> Table 1 shows the characteristics of the premature babies (lane 120)
Table 1 shows the characteristics of the premature babies.
- parameters: serum and >> parameters such as serum and (lane 133)
Renal function parameters such as serum and urinary concentration of creatinine, estimated GFR and urine output were normal in all babies
- Detecting early kidney damage >> Detecting kidney damage early (lane 154)
Detecting kidney damage early and with the precise location is currently a priority in the diagnosis of kidney diseases.
- There are only a few published study >> There are only a few published studies (lane 164)
There are only a few published studies on the determination of netrin-1 in newborns [8,17,18].
- significant higher >> significantly higher (lane 175)
- what is in agreement >> which is in agreement (lane 176)
In our studied groups eGFR was comparable in premature babies and statistically significant higher in term newborns, which is in agreement with the literature [19].
- the difference between both other groups was statistically significant >> it differs significantly from the other two groups (lane 184)
It turned out that we found the highest urinary level of netrin-1/cr. in babies born between 30-34 weeks of gestation and it differs significantly from the other two groups.
- and also urinary netrin-1/cr. >> or urinary netrin-1/cr. (lane 205)
We did not find any relationship between the concentration of urinary netrin-1 or urinary netrin-1/cr. and gender, delivery, birth weight, percentile of birth weight, Apgar score, prenatal steroid therapy, respiratory disorders (use of nCPAP, oxygen therapy), parenteral nutrition.
- further life >> further stages of life (lane 213)
Besides, it would be very valuable to conduct a long-term study to analyze the ne-trin-1/cr value especially in correlation with eGFR, in further stages of life.
6) Please review/re-write the following sentences; those are either too long and not clear or do not sound correct grammatically:
It allows to implement … (lane 155)
We changed according to Reviewer’s suggestion in Discussion
‘It allows to implement appropriate treatment with monitoring its effectiveness and prognosis, quickly and due to that increase the chance for cure [8].’
to this one:
‘It allows to quickly implement the appropriate treatment, monitor its effectiveness and prognosis, and due to that increase the chance for cure [8].’
Round 2
Reviewer 1 Report
The Authors satisfactorily answered the questions posed by the reviewers.